# Antibody responses induced by SHIV infection are more focused than those induced by soluble native HIV-1 envelope trimers in non-human primates

Jelle van Schooten[1][◐], Marlies M. van Haaren[1][◐], Hui Li[2], Laura E. McCoy[3,4], Colin Havenar-Daughton[5], Christopher A. Cottrell[6], Judith A. Burger[1], Patricia van der Woude[1], Leanne C. Helgers[1], Ilhan Tomris[1], Celia C. Labranche[7], David C. Montefiori[7], Andrew B. Ward[6,8,9], Dennis R. Burton[3,8,9,10], John P. Moore[11], Rogier W. Sanders[1,11], Shane Crotty[5], George M. Shaw[2], Marit J. van Gils[1]*

1 Department of Medical Microbiology, Amsterdam Infection & Immunity Institute, Amsterdam UMC, location AMC, University of Amsterdam, Amsterdam, The Netherlands, 2 Department of Medicine, University of Pennsylvania, Philadelphia, Pennsylvania, United States of America, 3 Department of Immunology and Microbiology, The Scripps Research Institute, La Jolla, California, United States of America, 4 Division of Infection and Immunity, University College London, London, United Kingdom, 5 Division of Vaccine Discovery, La Jolla Institute for Allergy and Immunology, La Jolla, California, United States of America, 6 Department of Integrative Structural and Computational Biology, The Scripps Research Institute, La Jolla, California, United States of America, 7 Laboratory for AIDS Vaccine Research and Development, Duke University Medical Center, Durham, North Carolina, United States of America, 8 International AIDS Vaccine Initiative—Neutralizing Antibody Center (IAVI-NAC), The Scripps Research Institute, La Jolla, California, United States of America, 9 Center for HIV/AIDS Vaccine Development (CHAVD), The Scripps Research Institute, La Jolla, California, United States of America, 10 Ragon Institute of MGH, MIT and Harvard, Cambridge, Massachusetts, United States of America, 11 Department of Microbiology and Immunology, Weill Medical College of Cornell University, New York, New York, United States of America

◐ These authors contributed equally to this work.
* M.J.vangils@amsterdamumc.nl

**Data Availability Statement:** The BG505-specific BCR sequences are deposited to DDBJ/ENA/GenBank with accession numbers MZ423305 -

## Abstract

The development of an effective human immunodeficiency virus (HIV-1) vaccine is a high global health priority. Soluble native-like HIV-1 envelope glycoprotein trimers (Env), including those based on the SOSIP design, have shown promise as vaccine candidates by inducing neutralizing antibody responses against the autologous virus in animal models. However, to overcome HIV-1's extreme diversity a vaccine needs to induce broadly neutralizing antibodies (bNAbs). Such bNAbs can protect non-human primates (NHPs) and humans from infection. The prototypic BG505 SOSIP.664 immunogen is based on the BG505 *env* sequence isolated from an HIV-1-infected infant from Kenya who developed a bNAb response. Studying bNAb development during natural HIV-1 infection can inform vaccine design, however, it is unclear to what extent vaccine-induced antibody responses to Env are comparable to those induced by natural infection. Here, we compared Env antibody responses in BG505 SOSIP-immunized NHPs with those in BG505 SHIV-infected NHPs, by analyzing monoclonal antibodies (mAbs). We observed three major differences between BG505 SOSIP immunization and BG505 SHIV infection. First, SHIV infection resulted in more clonal expansion and less antibody diversity compared to SOSIP immunization, likely

MZ423381 and MZ423382 - MZ423458. The accession numbers for the negative stain 3D EM reconstructions in the Electron Microscopy Data Bank are EMD-24243, EMD-24245, EMD-24246, EMD-24244, EMD-24247.

**Funding:** This work was supported by the HIV Vaccine Research and Design (HIVRAD) program (P01 AI110657) (A.B.W., R.W.S, J.P.M.), NIH CHAVI-ID (UM1 AI100663) and CHAVD (UM1 AI44462) awards (A.B.W., D.R.B.), NIH R01 AI13082 (J.P.M.), the International AIDS Vaccine Initiative Neutralizing Antibody Center, the Bill and Melinda Gates Foundation CAVD (OPP1115782, OPP1132237, OPP1084519, OPP119635, INV-002022), and the European Union's Horizon 2020 research and innovation program under grant agreement no. 681137 (R.W.S.). R.W.S. is a recipient of a Vici fellowship from the Netherlands Organization for Scientific Research (NWO). M.J.v. G. is supported by an amfAR Mathilde Krim Fellowships in Basic Biomedical Research grant number 109514-61-RKVA and the 2017 AMC Fellowship. J.v.S is a recipient of a 2017 AMC Ph. D. Scholarship. The funders had no role in study design, data collection and analysis, decision to publish, or preparation of the manuscript.

**Competing interests:** The authors have declared that no competing interests exist.

because of higher and/or prolonged antigenic stimulation and increased antigen diversity during infection. Second, while we retrieved comparatively fewer neutralizing mAbs (NAbs) from SOSIP-immunized animals, these NAbs targeted more diverse epitopes compared to NAbs from SHIV-infected animals. However, none of the NAbs, either elicited by vaccination or infection, showed any breadth. Finally, SOSIP immunization elicited antibodies against the base of the trimer, while infection did not, consistent with the base being placed onto the virus membrane in the latter setting. Together these data provide new insights into the antibody response against BG505 Env during infection and immunization and limitations that need to be overcome to induce better responses after vaccination.

## Author summary

A vaccine against HIV-1 would present a major breakthrough in the fight against HIV/AIDS. However, HIV-1 diversity, in particular in the envelope glycoproteins, proves a major hurdle for HIV-1 vaccine design. While broadly neutralizing antibodies develop to some degree in 20–30% of HIV-1-infected individuals and can protect non-human primates (NHPs) from virus infection, experimental HIV-1 vaccines have so far been unable to consistently induce such antibodies. A few years ago, soluble native-like HIV-1 envelope trimers, including SOSIP trimers, were developed which enabled the induction of neutralizing antibodies that could protect NHPs from infection with the sequence-matched virus. Here, we compared monoclonal antibodies from NHPs that were immunized with the SOSIP trimer or infected with a sequence-matched SHIV to better understand the successes and shortcomings of antibody development after SOSIP immunization compared to infection. Antibodies induced by infection were less diverse, but more clonally expanded and more potent in neutralizing the autologous virus. This is most likely a result of more and longer antigen stimulation and increased diversity of the envelope trimer during infection. Mimicking this extended antigen stimulation and variation with vaccination strategies might help to induce (broadly) neutralizing antibodies more efficiently.

## Introduction

A vaccine against the human immunodeficiency virus type I (HIV-1) will be critical to ending the HIV/AIDS pandemic. However, the immense HIV-1 diversity complicates the development of a vaccine against most circulating HIV-1 strains. Particularly, the enormous diversity in the envelope glycoproteins (Env), the targets for neutralizing antibodies (NAbs), hampers the induction of cross-reactive antibodies. Broadly neutralizing antibodies (bNAbs), i.e. NAbs that can neutralize a broad range of HIV-1 subtypes, develop to some degree in 20–30% of HIV-1 infected individuals [1] and, at appropriate titers, are able to protect non-human primates (NHPs) [2–4] from infection. However, no experimental HIV-1 vaccine has been able to consistently induce bNAb responses yet. Recombinant native-like HIV-1 Env trimers, including those based on the SOSIP design, are currently exploited in vaccination strategies aimed to induce bNAbs. Over the past few years, various SOSIP trimers with improved immunogenicity relative to the original design have been developed [5,6] based on the BG505 [7] and other HIV-1 *env* sequences [8,9]. The BG505 *env* sequence is of particular interest because serological analyses demonstrated that the BG505 HIV-1-infected infant developed bNAb

responses. Whether this was mediated by a diverse polyclonal response or a single bNAb lineage remains unknown [10,11]. Longitudinal analysis of BG505 *env* has revealed that the V1V2 region and a 10-residue stretch of gp120's C3 were among the regions under selective pressure in the HIV-infected infant from whom the BG505 virus was isolated [11]. Because of the limited availability of material from the BG505 HIV-1-infected infant many gaps remain in our understanding of BG505 infection and the ensuing antibody responses.

Various immunization studies have been performed with BG505 SOSIP immunogens aimed to induce similar bNAb responses as those developed in the BG505 HIV-1-infected infant. Immunization with BG505 SOSIP induced bNAbs in cows [12], but failed to do so in other animal models, including NHPs which have an antibody repertoire that resembles the human antibody repertoire more closely [11,13–17]. Nonetheless, BG505 and other SOSIP trimers have consistently induced autologous Tier-2 NAbs in rabbits and NHPs [11,13–17] and, in the latter case, the NAbs conferred protection against infection [17]. These NAb responses are generally strain-specific although some neutralization breadth has been observed in BG505 SOSIP-immunized NHPs, especially in those with high neutralization titers against the autologous virus [15].

The BG505 *env* sequence lacks three N-linked glycosylation sites (PNGS) that are conserved in most isolates [18–23]. Two of these are located close together and create a large strain-specific 241/289 glycan hole (HXB2 amino acid numbering). Extensive serological analyses and the isolation of monoclonal antibodies (mAbs) from BG505 SOSIP-immunized rabbits, guinea pigs, and NHPs demonstrated the immunodominance of this glycan hole after vaccination [18,20,21,24], explaining the narrow breadth of these antibody responses. Furthermore, the lack of the N465 PNGS resulted in another immunodominant area on the BG505 Env trimer [19,24–27]. This PNGS is located near the 10-residue stretch in the C3 region that was under selective pressure in the BG505 HIV-1-infected infant [11], hence referred to as the C3/V5 epitope. This C3/V5 epitope appeared to be more immunogenic in NHPs compared to rabbits [19,20,22]. Other previously identified targets for autologous NAbs on BG505 SOSIP include the gp41/gp120 interface, a region surrounding the glycan at residue 611, and an epitope near the apex of the trimer, involving the V1 region [19,21–24,27].

BG505 SOSIP immunization also induces Tier-1 and non-NAb responses directed to regions such as the V3 and the base of the soluble trimer, respectively [5,13,20–22,28]. These immunodominant epitopes have been postulated to distract the immune system from developing antibody responses to the more desired NAb epitopes due to an unfavorable immunodominance hierarchy [29]. Efforts to reduce the immunogenicity of the V3 region and non-NAb epitopes have not yet significantly improved neutralization breadth of the antibody responses as the SOSIP trimer immunodominant base remained exposed [13,15,30–32]. The multivalent display of SOSIP trimers on nanoparticles decreases the immunogenicity of the base of the trimer [33,34]. However, nanoparticle display did not increase nor broaden the NAb responses in BG505 SOSIP-immunized rabbits despite the restricted accessibility of the immunodominant glycan hole on the nanoparticle [33], although it did improve immunogenicity of other SOSIP trimers [33–35].

It is unclear whether these strain-specific NAb responses elicited by BG505 SOSIP could be used as a starting point to guide NAb responses towards neutralization breadth or that they need to be considered as "dead-end" responses that cannot be broadened. Co-evolution of BG505 Env and the bNAb response in the BG505 HIV-1-infected infant could provide information on how BG505 SOSIP can elicit similar types of antibody responses as those seen during natural infection. However, due to limited sample availability from the BG505 virus-infected infant these studies were not performed in detail [10].

Recent innovations have allowed the generation of infectious chimeric simian-human immunodeficiency virus (SHIV) with Envs from primary Tier-2 viruses, including BG505 [36,

37]. Electron microscopy-based polyclonal epitope mapping with sera from BG505 SHIV infected NHPs showed that the C3/V5 and the strain-specific 241/289 glycan hole, but not base binding antibodies are elicited in BG505 SHIV infected animals [22]. Here, we exploited the BG505 SHIV model and compared antibodies after BG505 SOSIP vaccination (SOSIP-mAbs) with those induced by BG505 SHIV infection (SHIV-mAbs) at the monoclonal antibody level. We found that SOSIP-mAbs were less clonally expanded and more diverse in terms of their Ig gene usage and epitope specificity compared to SHIV-mAbs. In contrast, only a few SHIV-mAb lineages were isolated, all of which underwent extensive clonal expansion. These lineages targeted either the 241/289 glycan hole or, more often, the trimer apex. The latter responses were the most potent NAb responses identified and focused around the V1 region. Antibodies against the 241/289 strain-specific glycan hole were elicited in both immunized and infected NHPs, but these mAbs did not neutralize the autologous virus potently. Lastly, the majority of the SOSIP-mAbs (66%) targeted the base of the trimer and were unable to neutralize the autologous virus. In contrast, base-targeting mAbs were not elicited during SHIV infection as the base of the trimer is embedded in the viral membrane of BG505 SHIV. These data demonstrate that BG505 SOSIP immunization induces a somewhat different and more complex antibody response compared to BG505 SHIV infection, which was more focused, while important similarities were also observed.

## Results

### BG505 SHIV infection induces memory B cells against the N241/N289 glycan hole and other epitopes

We selected BG505-specific B cells from peripheral blood mononuclear cells (PBMCs) from two BG505 SOSIP.v5.2-immunized NHPs with strong neutralization titers, i.e. $ID_{50}$ titers of 863 and 910 for NHPs 99–12 and ROp15, respectively [14]. B cells were sorted from PBMCs collected at week 20, two weeks after the fourth BG505 SOSIP immunization. In addition, we sorted BG505-specific B cells from three NHPs 24 weeks after infection with BG505 SHIV that developed autologous neutralization titers [36]. The $ID_{50}$ titers were 769, 333, and 52 for NHPs 6454, 43335, and 6446, respectively and none of the selected NHPs had developed any neutralization breadth in their sera at the time of PBMC isolation. For some analyses, we also included the information from mAbs that were previously isolated from two BG505 SOSIP.664-immunized NHPs (rh1987 and rh2011) [21] to increase the comparison between mAbs induced by vaccination and infection.

For the immunized NHPs, BG505-specific B cells were selected by fluorescence-activated cell sorting (FACS) using two fluorescently labeled BG505 SOSIP proteins (S1 Fig). As a result, we specifically sorted antibodies against the closed, native-like conformation of Env and not those directed to aberrant forms of Env. For sorting of BG505-specific B cells from the BG505 SHIV-infected NHPs, we used BG505 SOSIP and BG505 SOSIP S241N probes. The latter probe contains a PNGS at position 241 (S1 Fig). This sorting strategy merely provided additional information about the proportion of selected B cells directed to the 241-glycan hole on BG505 SOSIP, but did not select for additional B cells compared to the sorting strategy used for the immunized NHPs. B cells specific for the BG505 SOSIP and/or BG505 S241N trimer were selected for mAb isolation. All infected NHPs showed similar binding patterns with, on average, 74% of the sorted B cells binding both BG505 SOSIP and its S241N variant (S1 Fig). The remaining specific B cells bound exclusively to BG505 SOSIP and not to the S241N variant, suggesting that the B cell receptor (BCR) of these cells targeted the 241 glycan hole.

## BG505 SHIV infection induces stronger clonal expansion of B cells compared to BG505 SOSIP immunization

We isolated 32 mAbs from the two BG505 SOSIP-immunized NHPs and 45 from the three BG505 SHIV-infected NHPs, all with ELISA-confirmed binding to BG505 SOSIP (Fig 1A). The heavy chain CDR3 (CDRH3) is often elongated in HIV-1 bNAbs to facilitate penetration through the glycan shield of Env. From NHP ROp15, we isolated 15 SOSIP-mAbs with a mean CDRH3 length of 17 amino acids (aa) (range: 7–22) (Fig 1B). The 17 SOSIP-mAbs isolated from NHP 99–12 had a mean CDRH3 length of 14 aa (range: 9–20). From the BG505 SHIV-infected NHPs 6454, 4335 and 6446, we respectively isolated 25, 6, and 14 SHIV-mAbs. These SHIV-mAbs had a mean CDRH3 length of 24 (range: 10–29), 14 (range: 10–16), and 11 (range: 9–16) aa for NHPs 6454, 43335, and 6446, respectively. The CDRH3 length of SOSIP-mAbs varied substantially more than that of SHIV-mAbs. While mAbs from animal 6454 had substantially longer CDRH3 domains, the CDRH3 lengths of the mAbs from the other four animals were similar to those found for circulating B cells in rhesus macaques (14 aa on average) [38].

We then performed a phylogenetic analysis using the heavy chain (HC) variable region sequences to assess the evolutionary relationship between the mAbs derived from the same NHP and to compare the sequence variability between the different NHPs (Fig 1C). The

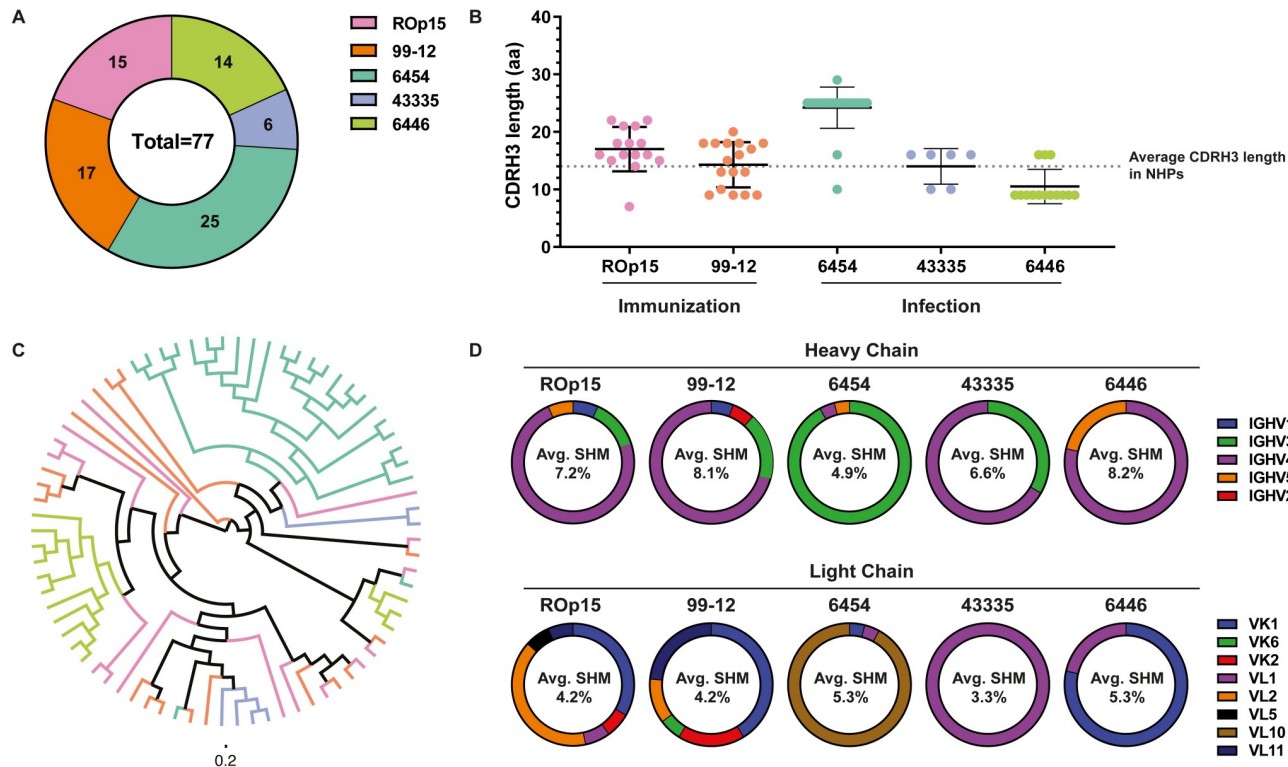

**Fig 1. Isolated mAbs induced by BG505 SOSIP are more clonally divergent than those induced by BG505 SHIV.** (A) In total, we successfully expressed 32 and 45 mAbs from two BG505 SOSIP (ROp15 and 99–12) and three BG505 SHIV-infected (6453, 43335, and 6446) NHPs, respectively. (B) CDRH3 amino acid length of individual mAbs isolated from the BG505 SOSIP-immunized and BG505 SHIV-infected NHPs. (C) Phylogenetic analysis using the heavy chain VDJ sequences to assess the evolutionary relationship between the mAbs and to compare the sequence variability between the different NHPs. Similar colors are used as panel A and B to distinguish the mAbs per NHP. (D) Variable gene usage for the heavy and light chains of the isolated mAbs. Average amount of somatic hypermutation (SHM) is indicated for each of the NHPs. SHM is determined as the percentage of nucleotide differences with a germline VH, VL, or VK gene segment.

immunoglobulin genes of the sorted cells from BG505 SOSIP-immunized NHPs were much more diverse whereas the majority of sequences derived from the BG505 SHIV-infected NHPs clustered together. The divergence in the sequences of the SOSIP-mAbs suggest that these originate from a variety of B cell precursors with limited to no clonal expansion (S1 Table). In contrast, the mAbs from the BG505 SHIV-infected NHPs likely originated from only one or two B cell precursors (Fig 1C and S1 Table).

Gene/allele usage and V-gene somatic hypermutation (SHM) were determined using an improved Indian origin rhesus macaque BCR germline database [21]. The majority of the SOSIP-mAbs from both ROp15 and 99–12 utilized HC variable genes from the VH4 family with an average SHM in the HC of 7.2% and 8.1%, respectively (Fig 1D), which was very comparable to the previously isolated mAbs from BG505 SOSIP immunized NHPs 1987 and 2011 [21]. ROp15 predominantly utilized kappa chain variable genes from the VK1 family and lambda chain variable genes from the VL2 family with an average SHM in the light chain (LC) of 4.2%. NHP 99–12 also used the VK1 family predominantly in its kappa chain variable genes and the VL11 family in its lambda chain variable genes. The average SHM in the LC for NHP 99–12 was also 4.2%.

Most of the SHIV-mAbs isolated from NHP 6454 belonged to one lineage and utilized the VH3 family variable genes in its HC and the VL10 family variable genes in its lambda chain (Fig 1D). The average SHM in NHP 6454 was 4.9% and 5.3% in its HC and LC, respectively. The two mAb lineages of NHP 43335 used the HC variable genes of the VH3 and VH4 families with an average SHM of 6.6%. The lambda chain variable genes of these two lineages both belonged to the VL1 family with an average SHM of 3.3%. The two lineages of NHP 6446 utilized either a kappa or lambda chain with respectively HC variable genes from the VH4 and VH5 families and LC variable genes from the VK1 and VL1 families. The SHM in the HC of NHP 6446 was 8.2% and 5.3% in its LC. Taken together, the data shows that the isolated mAbs both after immunization or infection predominantly use the VH3 and VH4 V-gene families and that there are no significant differences in the SHM levels after immunization or infection in either the HC or LC. We note that these estimates of SHM are not precise because of incompleteness of the rhesus macaque BCR germline database [21].

## Both BG505 SOSIP immunization and BG505 SHIV infection induce autologous NAbs

Next, the neutralizing activity of the isolated BG505-specific mAbs was tested against the autologous BG505 T332N virus (from here on referred to as BG505 virus). From every NHP at least one or more mAbs neutralized the autologous BG505 virus (Fig 2A). 27% (4 SOSIP-NAbs) and 6% (1 SOSIP-NAb) of the mAbs from NHPs ROp15 and 99–12, respectively, neutralized the autologous virus. The CDRH3 length of these 5 SOSIP-NAbs ranged from 7–21 aa, indicating no relationship between CDRH3 length and neutralization of the autologous virus. The percentage of isolated NAbs in this study is comparable to the 12–16% of NAbs from BG505 SOSIP.664-immunized NHPs 1987 and 2011 in a previous study (Fig 2A) [21]. From the SHIV-infected NHPs the proportion of isolated mAbs that neutralized the BG505 virus (SHIV-NAbs) was 92% (23 SHIV-NAbs), 33% (2 SHIV-NAbs) and 14% (2 SHIV-NAbs) from NHPs 6454, 43335, and 6446, respectively. The CDRH3 length of the SHIV-NAbs ranged from 10–25 aa, demonstrating no relationship between neutralization of the autologous virus and the CDRH3 length of these mAbs. NAbs isolated from the BG505 SHIV-infected NHPs in each individual belonged to only one antibody lineage, whilst the four NAbs from the BG505 SOSIP recipient ROp15 were clonally unrelated.

The $IC_{50}$ values of SOSIP-NAbs ranged from 0.12 to 21 µg/ml (Fig 2B). In contrast, the NAbs from the SHIV-NAb lineages RM54B and RM35A showed remarkable potency against

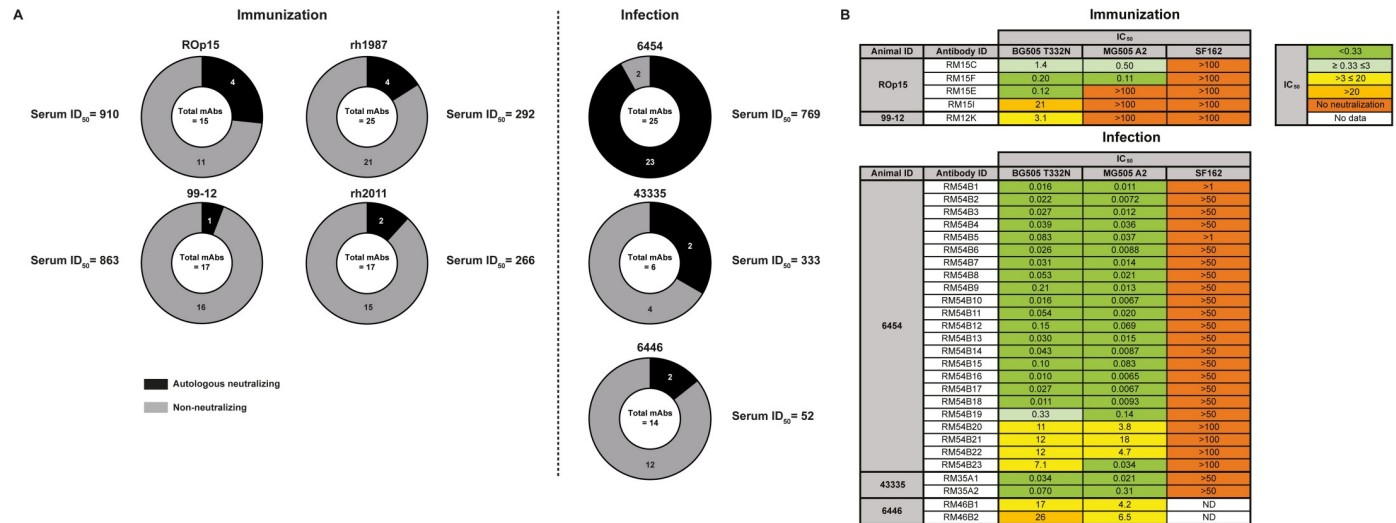

**Fig 2. Autologous NAbs isolated from both the BG505 SOSIP-immunized and BG505 SHIV-infected NHPs.** (A) Pie-charts indicating the amount of isolated mAbs able to neutralize the autologous BG505 T332N virus per NHP. We also included the information from mAbs that were previously isolated from two BG505 SOSIP.664-immunized NHPs (rh1987 and rh2011). The inhibitory dilution ($ID_{50}$) value next to the pie-chart indicates the autologous NAb titer of the serum at week 20 and week 24 post immunization and infection, respectively. (B) The half maximal inhibitory concentration ($IC_{50}$) values by which the NAbs neutralize the autologous BG505 T332N, closely-related MG505 A2, and heterologous SF162 viruses.

the BG505 virus with the majority of lineage members having $IC_{50}$ values below 0.10 μg/ml. Lineage members RM54B19-RM54B23 were less potent with $IC_{50}$ ranging from 0.33–12 μg/ml. SHIV-NAbs from NHP 6446 were the least potent SHIV-NAbs with $IC_{50}$ values higher than 15 μg/ml. We observed a relation between the serum neutralization titers and the number and potency of the isolated SHIV-NAbs. NHPs 6454 and 43335, which had high serum neutralization titers, yielded very potent NAb lineages whereas the NAb lineages isolated from NHP 6446, which had a lower serum $ID_{50}$, were much less potent (Fig 2A and 2B).

We also tested the SOSIP- and SHIV-NAbs against the related MG505 A2 virus (Fig 2B), which was isolated from the mother of the BG505 HIV-1-infected infant. It differs from the BG505 virus by 13 amino acids, including at position 241, where N241 creates a PNGS in MG505 A2 [7]. The majority of SOSIP and SHIV-NAbs neutralized the MG505 A2 virus with similar potencies compared to the BG505 virus. However, three SOSIP-NAbs were unable to neutralize MG505 A2 and two of these mAbs were later shown to target the region surrounding the glycan hole at S241 (see below). None of the tested SOSIP- or SHIV-mAbs were able to cross-neutralize the clade B Tier 1A virus SF162. This is consistent with neutralization of SF162 being dominated by V3-specificities that do not neutralize the Tier 2 virus BG505.

## Both SHIV infection and SOSIP immunization elicited mAbs that target the BG505 strain-specific 241/289 glycan hole

A very immunodominant epitope on the BG505 Env trimer is the 241/289 strain-specific glycan hole (Fig 3A) [11,18,19,21]. We tested the ability of the mAbs to neutralize and/or bind glycan variants that were created to fill the 241/289 strain-specific glycan hole of BG505 to determine the extent of this response. The mutant viruses and SOSIP proteins included S241N, knocking-in a PNGS at N241; P291T, knocking in a PNGS at N289, as well as the combination mutant S241N/P291T (S2 Fig and S2 Table). We identified 23 mAbs that targeted the 241/289 glycan hole, four of which were derived from BG505 SOSIP-immunized NHPs and 19 from SHIV-infected NHPs (S2 Table and S3 Fig). Three out of these 23 mAbs neutralized the

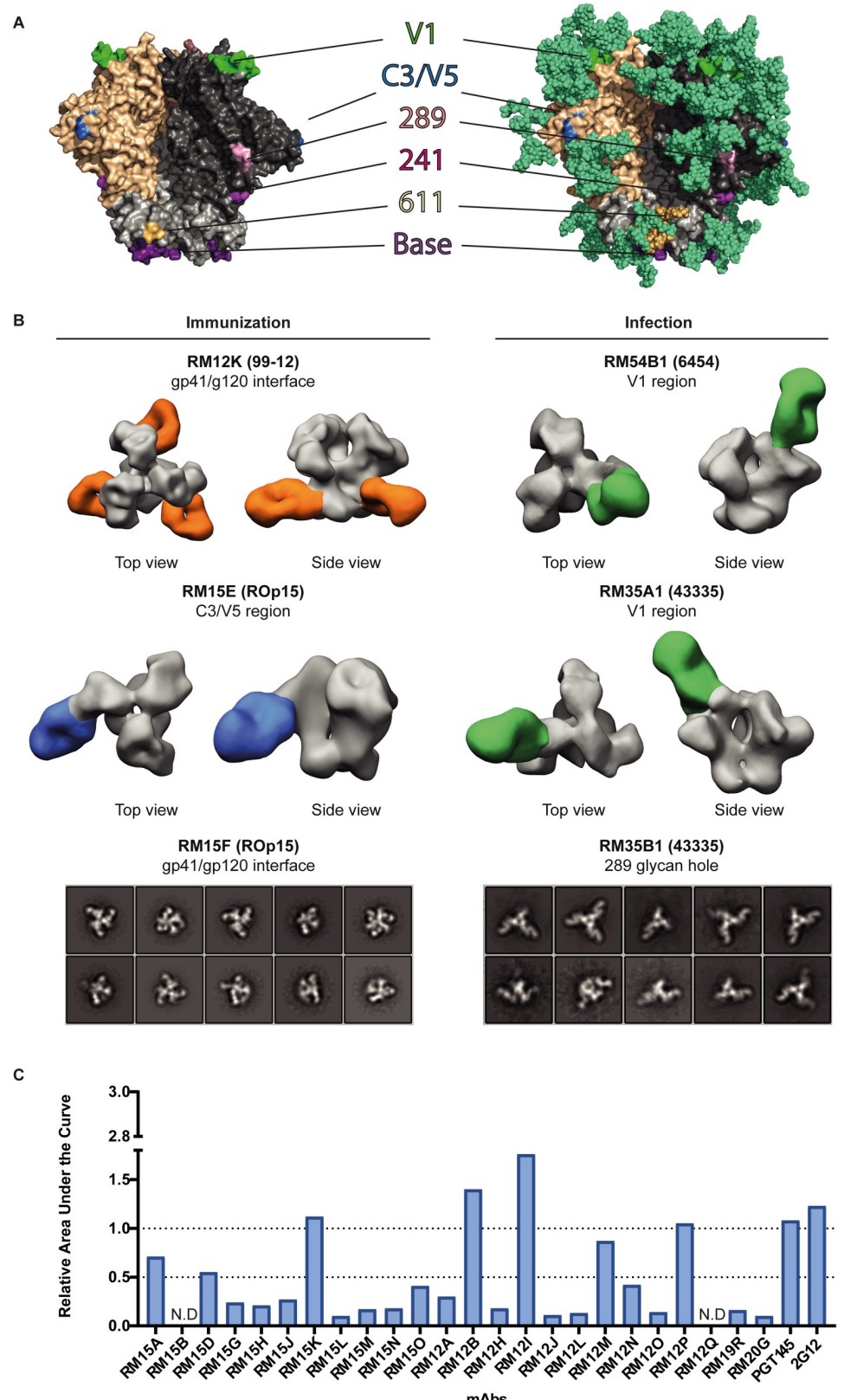

**Fig 3. Epitope mapping of mAbs isolated from BG505 SOSIP-immunized and BG505 SHIV-infected NHPs.** (A) Model of BG505 SOSIP without (left) and with (right) glycans. The different epitopes targeted by mAbs induced by

BG505 SOSIP are indicated in different colors. (B) Negative stain-electron microscopy was performed to verify various epitope specificities of the isolated mAbs. MAbs are shown in complex with BG505 SOSIP. For mAbs RM12K (orange), RM54B1 (green), RM15E (blue), and RM35A1 (green) 3D reconstructions are displayed. BG505 SOSIP is displayed in grey. For mAbs RM15F and RM35B1 the 2D class averages are shown. (C) ELISA binding of non-neutralizing mAbs from BG505 SOSIP-immunized NHPs to BG505 SOSIP variant (R500A+Q658K) relative to BG505 SOSIP. The BG505 SOSIP variant (R500A+Q658K) diminishes binding of antibodies to the base of the trimer. Value on the y-axis indicates the fold difference in area under the curve (AUC) of binding to BG505 SOSIP variant (R500A+Q658K) relative to the AUC of BG505 SOSIP.

autologous BG505 virus, one from the immunized NHP ROp15 (RM15I) and two from the infected NHP 6446 (RM46B1 and RM46B2). SOSIP-NAb RM15I was unable to neutralize the S241N, P291T, S241N + P291T BG505, and MG505 viruses (Fig 2 and S2 Fig). In addition, RM15I showed decreased binding to BG505 S241N in ELISA indicating that RM15I targets an epitope near the missing glycan at position 241 (S2 Table). RM46B1 and RM46B2 were also sensitive to substitution of residues 241 and 291 but could neutralize MG505 (Fig 2B). However, these mAbs were also sensitive to substitutions elsewhere (see below), complicating the interpretation of the data and pinpointing the exact epitope.

The remaining 20 mAbs that targeted the 241/289 glycan hole were non-neutralizing. Three of these non-NAbs were isolated from the immunized NHP 99–12, whereas the remainder were isolated from the BG505 SHIV-infected NHPs. NS-EM confirmed the 241/289 glycan hole specificity of the non-neutralizing SHIV-mAb RM35B1 (Fig 3B). In general, the 241/289 targeting mAbs were more sensitive to the P291T substitution while a minority were more or equally sensitive to the S241N substitution, pointing at differences in the fine-specificities within the 241/289 glycan hole epitopes (S2 Table and S2 Fig). We observed that both non-NAbs and NAbs targeting the strain-specific 241/289 glycan hole used either the VH4 or VH5 family genes in their HC.

## BG505 SOSIP immunization induces mAbs against a greater variety of epitopes compared to BG505 SHIV infection

Other epitopes frequently targeted after BG505 SOSIP immunization include the glycan hole at N611 caused by incomplete occupancy of the N611 PNGS, an epitope around C3 and V5 (termed C3/V5), the gp41/gp120 interface and, an epitope in the V1 region (Fig 3A) [11,18,19,21–23,26]. Only one NAb, RM15E derived from a SOSIP immunized animal, was clearly directed to the C3/V5 epitope (S2 Table and S2 Fig), as RM15E neutralization was greatly affected by the T465N mutation (S2 Fig). NS-EM imaging confirmed that RM15E targeted the C3/V5 region (Fig 3B). RM15E could not neutralize the MG505 A2 virus and neutralization was also affected by the P291T and S241N + P291T substitutions, although to a lesser extent, suggesting an interrelationship of the 241/289 and C3/V5 epitopes. In addition, mAb RM12K isolated from the immunized NHP 99–12 also showed reduced neutralization of BG505 T465N and was unable to neutralize MG505 A2. However, NS-EM imaging located the epitope of RM12K to the gp41/gp120 interface of BG505 SOSIP and not specifically the C3/V5 epitope (Fig 3B). Lastly, NAb RM54B19, derived from a SHIV-infected animal, could also not neutralize the T465N virus. However, based on the clonal relationship of RM54B19 to the V1-targeting antibody lineage, we mapped RM54B19 to the V1 and not the C3/V5 epitope (see below). Thus, the T465N substitution affects recognition of the C3/V5 epitope but also more distal epitopes including the V1 domain and the gp41/gp120 interface.

Incomplete glycan occupancy of the N611 PNGS on BG505 SOSIP has been linked to the induction of mAbs able to neutralize BG505 N611A but not the BG505 virus [21]. Indeed, it was shown that the N611 PNGS on BG505 SOSIP is occupied in only ~40% of the cases [39].

Moreover, six mAbs isolated from two BG505 SOSIP.664-immunized NHPs in a previous study bound to an epitope near the N611 glycan [21]. In this study, we found only one mAb from the BG505 SOSIP-immunized NHP 99–12 (RM12F) that was able to potently neutralize BG505 N611A virus but not the parental BG505 virus. This suggests that RM12F targets a region that is shielded by the N611 glycan (S2 Fig). We did not find any SHIV-mAbs directed to the N611 region, which could be explained by full occupancy of this site on BG505 viral Env [40].

A region surrounding the V1 on BG505 SOSIP trimers previously induced NAbs in a proportion of BG505 SOSIP-immunized rabbits [19,23] and NAbs against this epitope have been associated with protection against autologous virus infection in NHPs [22]. Therefore, we created a BG505 virus and SOSIP variant that contained a single amino acid insertion in the V1 region near residue N133 that increases the length of the V1 and moves the N133-glycan site by one position (BG505 133aN) to eliminate binding of mAbs with this epitope specificity [19, 23]. SOSIP-NAb RM15C and 25 SHIV-NAbs were unable to neutralize this BG505 133aN virus (S2 Fig). Multiple members from the V1 region targeting SHIV-NAb lineage RM54B were also affected by other substitutions such as the T465N mutation (S2 Fig), but NS-EM confirmed the V1 epitope specificity for the representative SHIV-mAbs RM54B1 and RM35A1 (Fig 3B). Another SOSIP-NAb (RM15F) showed diminished neutralization potency against the BG505 133aN variant, but NS-EM identified the gp41/gp120 interface as its epitope (Fig 3B).

Taken together we observed that SOSIP-mAbs target a wide variety of epitopes on the BG505 Env trimer, including the 241/289-glycan hole, the N611 glycan hole, the C3/V5 domain, the gp41-gp120 interface, and the V1 region. In contrast, the SHIV-mAbs we generated recognized either the V1 region or the 241/289 glycan hole, but no other domains.

## BG505 SOSIP immunization induces antibodies targeting the trimer base while BG505 SHIV infection does not

We then studied the non-neutralizing mAbs that were induced by SOSIP immunization. The majority (75%) of the non-neutralizing SOSIP-mAbs was unable to bind the BG505 gp120 monomer in ELISA (S2 Table). In addition, many of these SOSIP-mAbs competed with two mAbs targeting the trimer base, RM19R and RM20G, that were previously isolated from BG505 SOSIP.664 immunized NHPs [21]. (S3A Fig). Furthermore, the binding of 17 out of 23 of these mAbs was reduced or abrogated by a combination of the R500A and Q658K substitutions that eliminated RM19R binding [21] (Fig 3C). Finally, the base specificity was visualized by negative-stain electron-microscopy (NS-EM) for SOSIP-mAb RM15A (S3B Fig). Overall, these results are in concordance with previous studies that identified the base of BG505 SOSIP to be highly immunodominant [20–22]. From these results we can conclude that a large proportion (66%) of the non-neutralizing responses elicited by BG505 SOSIP immunization are directed to the base of the Env trimer.

For the SHIV-mAbs, we demonstrated that all non-NAbs target the 241/289 glycan hole and we did not find any mAbs directed to the base of the trimer. This can simply be explained because the Env trimer is embedded in the viral membrane on BG505 SHIV viral particles excluding the base as a target for antibodies.

## V1-targeting NAbs are amongst the most potent NAbs isolated from both BG505 SHIV and BG505 SOSIP-immunized NHPs

The neutralization potencies of the SOSIP-NAbs and SHIV-NAbs were not significantly different (P = 0.44, unpaired t-test). However, we observed that the majority of the V1 directed

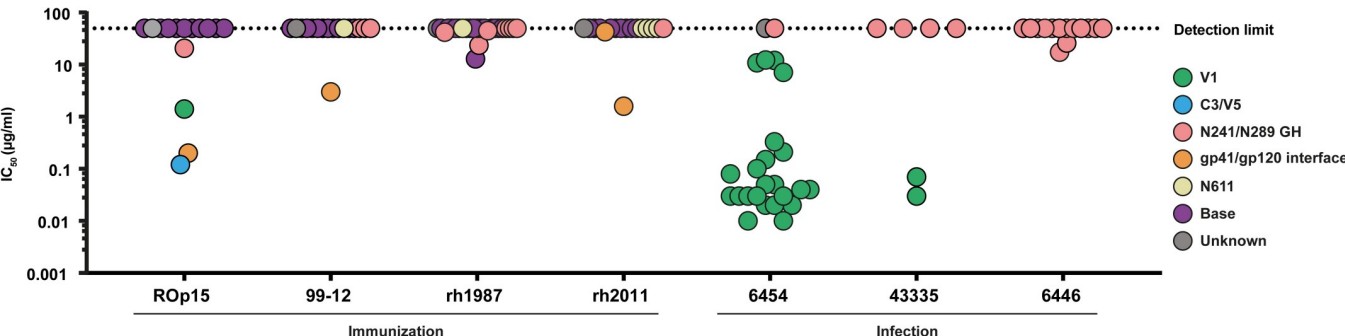

**Fig 4. Potency and epitope specificities of isolated mAbs.** Half maximal inhibitory concentration ($IC_{50}$) for every mAb isolated from the different NHPs. Epitopes targeted by these mAbs are indicated in different colors showing the relationship between neutralization potency and the various epitopes. Antibodies that have a potency higher than 50 µg/mL are classified as non-neutralizing. We also included the information from mAbs that were previously isolated from two BG505 SOSIP.664-immunized NHPs (rh1987 and rh2011).

NAbs did have a remarkably potency (Fig 4), with $IC_{50}$ values ranging from 0.010–0.33 µg/ml, while the V1 targeting SOSIP-NAb RM15C was somewhat less potent with an $IC_{50}$ value of 1.4 µg/ml. All V1 targeting NAbs used the VH3 family gene in their HC. Furthermore, we did not identify non-neutralizing mAbs against this domain suggesting that this epitope predominantly elicits NAbs. In contrast, the majority of the mAbs directed against the 241/289 glycan hole were non-NAbs. Moreover, both SHIV- and SOSIP-NAbs targeting the 241/289 glycan hole were remarkably less potent compared to the V1 targeting NAbs with $IC_{50}$ values above 15 µg/ml. SOSIP-NAbs targeting the C3/V5 epitope (RM15E) and the gp41/gp120 interface (RM15F and RM12K) neutralized BG505 with relatively low $IC_{50}$'s of 0.12 µg/ml, 0.20 µg/ml, and 3.1 µg/ml respectively (Fig 4). Furthermore, the neutralization potencies of the SOSIP and SHIV-NAbs were comparable to those of mAbs targeting similar epitopes isolated from BG505 SOSIP.664 immunized NHPs rh1987 and rh2011 in a previous study [21] (Fig 4).

## Selected NAbs do not neutralize heterologous viruses

Many of the NAbs cross-neutralized the closely related MG505 A2 virus derived from the mother of the BG505 HIV-1-infected infant (Fig 2B). To test the neutralization breadth of selected NAbs we used a panel of viruses representative of the global HIV-1 diversity [41] (S4 Fig). NAbs targeting the 241/289, C3/V5, and 611 epitopes were previously shown to lack neutralization activity against heterologous viruses since most of the viruses have N-linked glycans present on these positions [18,21]. We therefore did not include NAbs with such specificities in the analysis. In contrast, gp41/gp120 directed NAbs have the potential to neutralize heterologous viruses since this epitope includes the highly conserved fusion peptide (FP) [21,22]. Therefore, we assessed the neutralization breadth of the gp41/gp120 targeting SOSIP-mAbs RM12K and RM15F. In addition, the potent V1 targeting SOSIP-NAb RM15C and the SHIV--NAbs RM54B1 and RM35A2 were also tested for heterologous neutralization. None of these NAbs exhibited the ability to neutralize any viruses from the global panel (S4 Fig).

## Discussion

Since the first generation of BG505 SOSIP.664 trimers was developed, BG505 SOSIP.664 and subsequent variants have been extensively studied in various animals and humans [42–45]. In contrast, the bNAb response(s) that developed in the HIV-1-infected infant from whom the BG505 *env* sequence was isolated have been less well defined due to lack of sample availability. Characterization of the antibody responses developed during natural BG505 infection could

provide important insights into how to elicit bNAb responses by immunization. Therefore, we searched for an alternative model to recapitulate the course of BG505 HIV-1 Env and antibody co-evolution and compared it to antibody development after BG505 SOSIP immunization. To do this, we mapped the epitopes of mAbs isolated from two BG505 SOSIP.v5.2-immunized and three BG505 SHIV-infected NHPs, 20 and 24 weeks after immunization or infection, respectively. We observed that SOSIP-mAbs target a wide variety of epitopes on BG505 SOSIP such as the 241/289 and 465-glycan hole, the N611 region, the V1 region, gp120/gp41 interface and the base of the trimer comparable to observations made in previous studies [18,19,21–23,26]. In contrast, SHIV-mAbs exclusively target the regions surrounding the V1 or the 241/289-glycan hole. Longitudinal sequence analysis of Env in the BG505 SHIV-infected NHPs (Hui *et al.*, manuscript in preparation) revealed that the V1 and the 241/289 regions were indeed under selective pressure. Overall, these data show that immunization with soluble SOSIP trimers elicit mAbs targeting the lower part of the trimer compared to mAbs elicited during infection which are more focused to the top of the trimer.

A difference between BG505 SOSIP and SHIV is the presentation of Env to the immune system, i.e. BG505 SHIV does not include the stabilizing SOSIP mutations and is expressed on the viral membrane. The lack of SOSIP mutations in BG505 SHIV Env enables it to adopt a variety of conformations including aberrant forms of Env such as gp41 monomers and incorrectly processed gp41/gp120 oligomers. In contrast, BG505 SOSIP is stabilized in a prefusion native-like conformation, optimally presenting (b) NAb epitopes. As a result, BG505 SHIV also elicits antibodies against epitopes that are normally not accessible on native-like SOSIP trimers. In this study we excluded antibodies against these incorrectly processed forms of Env from our analysis by using a soluble SOSIP Env trimer to sort B cells. To what extent the antibody response in the BG505 SHIV infected NHPs is influenced by these aberrant conformations of Env we cannot say. In addition, anchoring of the trimer in the viral membrane on viral particles makes the lower part of the trimer much less accessible for antibodies. This could explain why the response in the BG505 immunized NHPs is more focused to the lower part of the trimer compared to antibodies elicited by BG505 SOSIP.

Base-targeting mAbs represented ~66% of the mAbs isolated from BG505 SOSIP-immunized NHPs, a finding consistent with results of other studies [21]. If anything, this might be an underestimation as the BG505 SOSIP trimers we used to sort the single BG505-specific B cells were non-covalently linked to fluorescent streptavidin at the base of the trimer, which could have hindered the binding of a selection of mAbs against this domain. The dominance of the base response likely distracts the immune system from maturing antibody responses to neutralizing epitopes on the trimer [28,29,46,47]. In addition, base-directed antibody responses induce disassembly of soluble Env trimers into protomers that expose additional off-target epitopes present inside the native trimer [28]. The base of soluble Env trimers should be modified to restrict antibody responses to this region. Various strategies are currently being pursued to hide the base of SOSIP trimers such as using nanoparticle display [33–35] or glycosylation of the base [31]. Both strategies reduced base-directed antibody responses *in vivo* [31,34].

Klasse *et al.* (2018) previously described that neutralization by the sera from BG505 SOSIP-immunized NHPs ROp15 and 99–12 was greatly affected by knocking-in the N465 PNGS [19]. Additionally, electron microscopy polyclonal epitope mapping (EMPEM) of sera from BG505 SOSIP.664 trimer immunized NHPs has demonstrated that C3/V5 targeting NAbs develop in NHPs with high autologous neutralization titers, but that the presence of V1/V3 NAbs was required for durable protection against repetitive SHIV challenges [22]. Consistent with these observations, we isolated NAbs targeting both the C3/V5 and V1 region from NHP ROp15 with a high neutralizing serum titer against the autologous virus. From NHP 99–12, we only

isolated one SOSIP-NAb (RM12K) directed to the gp41/gp120 interface, however neutralization was also affected by knock-in of the N465 PNGS, which could explain the similar effects of the N465 PNGS knock-in on the serum neutralization of NHP 99–12 [19]. EMPEM showed that two NHPs immunized with BG505 SOSIP.664 trimers did not develop any mAbs targeting the C3/V5 region [21], while sera of these NHPs was greatly affected by the knock-in of the N465 PNGS [19]. This suggests that introduction of the 465 glycan can also affect neutralization of mAbs directed towards other epitopes such as the gp41/gp120 interface and 241/289 glycan hole.

bNAbs against the FP of Env have been isolated from various HIV-1 infected individuals [48–50]. The FP is a key functional site of HIV-1 and is highly conserved between viral subtypes, making it an attractive target for vaccine design strategies. Previous immunization studies with BG505 SOSIP have elicited mAbs against the FP but lacked neutralization breadth due to dependency on poorly conserved residues and the inability to accommodate the N611 PNGS [21,22]. In this study, we isolated two SOSIP-NAbs against the gp41/gp120 interface that might include the FP in their epitope. These two SOSIP-NAbs did not show any neutralization breadth, possibly due the dependency on poorly conserved residues as neutralization of BG505 was not enhanced by the deletion of the N611 PNGS. In contrast, we did not isolate any SHIV-mAbs directed to the FP, consistent with other studies [22]. This might be the result of differences in occupancy of the N611 PNGS between BG505 SHIV and BG505 SOSIP Env [39]. However, FP-directed immunization strategies were able to elicit moderately broad NAbs in some animals, which demonstrates the potential of the FP as a vaccine target [51–53]. Strategies aimed to elicit FP-targeting responses have deleted the N611 PNGS to facilitate easier initial access to the FP. Additional boosts with fully glycosylated Env trimers are then probably needed to steer antibody maturation to accommodate the conserved N611 PNGS present on circulating HIV-1 viruses [51–53]. Gradually re-introducing glycans on Env immunogens could also be applied to other epitopes then the FP, i.e. this strategy has been successful to elicit a bNAb against the CD4 binding-site in a rabbit immunized with a Env NFL-liposome immunogen regiment [54].

We also found that SHIV-mAbs are more clonally expanded indicated by the extensive clonal families that we isolated from the SHIV-infected NHPs. Whereas mAbs isolated from SOSIP-immunized NHPs were mostly clonally unrelated. This is likely due to the persistent availability of the continuously evolving BG505 Env during infection [36]. This presumably results in extensive clonal expansion in the germinal centers leading to a more focused immune response. In contrast, after immunization, soluble BG505 SOSIP is degraded over time and therefore, the immune system is exposed to BG505 SOSIP for a shorter period of time. In addition, the BG505 SOSIP immunogen does not present sequence diversity. As a consequence of these two factors, germinal center development might be restricted after immunization, resulting in limited clonal expansion. Indeed, extended immunogen release using osmotic pumps enhanced germinal center activity and increased autologous Tier-2 NAb serum titers compared to conventional immunization in BG505 SOSIP-immunized NHPs [15]. Moreover, in another study it was also shown that slow-delivery of Env immunogens improved germinal center and Env-specific T follicular helper ($T_{FH}$) cell responses, although the concomitant antibody responses remained narrow in specificity [55]. The combination of increased antigen stimulation and antigen variation might help to enhance germinal center responses, eventually leading to more broad and potent antibody responses.

All SOSIP-NAbs and SHIV-NAbs directed to the V1 region were able to potently neutralize BG505. However, these NAbs never developed into bNAbs during the time frame of these studies. High resolution cryo-EM studies of V1-targeting NAbs isolated from both BG505 SOSIP-immunized rabbits and NHPs have demonstrated that these NAbs primarily contact

the highly variable V1 loop [22,23]. Therefore, these NAbs likely do not have the potential to develop into bNAbs and will most likely interfere with the development of bNAbs against the N332 supersite. This might explain why the V1-targeting NAbs isolated in this study do not neutralize any heterologous viruses as they likely contact the variable V1 loop. To broaden the antibody response, additional engineering of BG505 SOSIP is required to steer the antibody responses away from these strains-specific V1-targeting NAbs to allow the development of bNAbs to the more-desired N332 supersite. The high potency of these V1-targeting NAbs compared to NAbs targeting other epitopes might be related to the angle these NAbs use to approach the trimer. MAbs that target epitopes located lower on the trimer might need a specific angle of approach to cope with the viral membrane whereas mAbs targeting the apex of the trimer can be more flexible in terms of their approach angle.

In summary, the data shows that immunizing NHPs with BG505 SOSIP induces a wide array of mAb lineages with varying specificities, whereas the antibody response during BG505 SHIV infection is more focused and targets epitopes higher up on the Env trimer. This data suggests that future immunization studies should try to focus the antibody responses towards one specific conserved epitope on BG505 SOSIP to increase clonal expansion and potentially the breadth of the antibody response. The combination of state-of-the-art immunogen design strategies such as nanoparticle display and slow-delivery of glycan-modified Env immunogens could help to focus the antibody response towards conserved sites on the Env trimer. The V1 region targeting NAbs that we isolated abundantly from BG505 SHIV-infected NHPs, but more scarcely from the BG505 SOSIP-immunized NHPs will, despite their potency, cannot develop further into bNAbs. Therefore, the V1 is likely not a good target to focus vaccine design on. In contrast, responses to the gp41/gp120 interface and FP, and possibly others, might be more amendable to broadening [51–53]. Together these data indicate that immunization strategies with a homogenous antigen are not sufficient to drive affinity maturation towards neutralization breadth. Therefore, the development of improved innovative immunization strategies is required to increase clonal expansion and the development of bNAb lineages i.e. targeting more conserved sites.

## Materials and methods

### Ethics statement

PBMCs in this study were obtained from animal studies that have been described previously [14, 36]. These studies were carried out according to the NIH guidelines in compliance with IACUC regulations. The rhesus macaques were housed, immunized and bled in compliance with the Animal Welfare Act and other federal statutes and regulations relating to animals, and in adherence to the Guide for the Care and Use of Laboratory Animals, National Research Council, 1996.

### Animals, immunization and infection

The PBMCs of ROp15 and 99–12 were obtained from a previous immunization study described in Havenar-Daughton *et al*. 2016 [14]. In brief, two Indian rhesus macaques received 100 μg doses of BG505 SOSIP.v5.2 trimers formulated in PLGA (MPL+R848) at a 6 week interval. Each immunization was delivered subcutaneous in the medial right and left mid-thigh. Blood was drawn two weeks after the fourth BG505 SOSIP.v5.2 trimer immunization and used for the isolation PBMCs. Both ROp15 and 99–12 were housed at the Yerkes National Primate Research Center.

The PBMCs of 6454, 43335 and 6446 were obtained from BG505 SHIV-infected NHPs described in Hui *et al*. 2016 [36]. In brief, Indian rhesus macaques 6454 and 43335 were

infected with a mixed viral population of SHIV.A.BG505.332T and SHIV.A.BG505.332N whereas Indian rhesus macaque 6446 received the SHIV.A.BG505.332T variant. All of these SHIV.A.BG505.332(N/T) variants contained either a serine, methionine, tyrosine, histidine, tryptophan, or phenylalanine at position 375. Blood was drawn 24 weeks post infection and used for the isolation of PBMCs.

## SOSIP trimer and virus production

SOSIPv5.2 trimers were produced as previously described [56]. In brief, 1 L HEK293F cells (Invitrogen, cat no. R79009) were diluted to a cell density of 1 million cells/ml in FreeStyle Medium (Life Technologies) one hour before transfection. Plasmids encoding for SOSIP and furin were co-transfected in a 4:1 ratio using 1 mg/ml Polyethylenimine Hydrochloride (PEI-max) (Polysciences Europe GmBH, Eppelheim, Germany) as the transfection agent. After 6 days, the cells were spun down and the supernatant was filtered using 0.22 μm pore size Steri-topes (Millipore, Amsterdam, The Netherlands). The supernatant was run overnight at 4°C (0.5–1.0 ml/min) over a PGT145 affinity column, prepared as previously described [9]. SOSIP trimers were eluted with 3 M MgCl$_2$ pH 7.2, 20 mM TrisHCl into an equal volume of TN75 buffer (20 mM TrisHCl pH 8.0, 75 mM NaCl). Buffer exchange into TN75 buffer was performed using Vivaspin20 centrifugal filters with 100 kDa MW cut-off (Sartorius, Göttingen, Germany).

Viruses were produced by transfection of virus plasmid DNA into HEK293T cells (ATCC, CRL-11268) using lipofectamin2000 (Invitrogen). Supernatant containing virus was harvested 3 days after transfection and used in neutralization assays.

## B cell selection and and mAb cloning

CD20+/viability- B cells were selected by fluorescence-activated cell sorting (FACS) using a bio-tinylated BG505 SOSIP.v5.2 and BG505 SOSIP.v5.2 S241N trimers conjugated to respectively strep-PE, strep-APC, or strep-AF647 and strep-PE (all Thermo Fisher Scientific). For the immunized NHPs, B cells were single-cell sorted that bound to two fluorescently labeled BG505 SOSIP.v5.2 trimers. For the infected NHPs, B cells that bound to BG505 SOSIP and/or the BG505 SOSIP S241N were single cell sorted. The sorted B cells were lysed in lysis buffer (RNAse inhibitor (40 U/μl) (Thermo Fisher Scientific), 5X First strand superscript III buffer (Invitrogen), 0.1 M DTT (Invitrogen), and MQ) and released mRNA was converted to cDNA by RT-PCR. Subsequently, 6 μL of RT-PCR reaction mixture (random hex primers 200 ng/μL (Thermo Fisher Scientific), dNTP mix 6 mM each (New England Biolabs), 50 U Superscript III RTase, and MQ) was added directly to the wells of the single cell sorted B cells. The following PCR program was performed: 42°C for 10 min, 25°C for 10 min, 50°C for 60 min and 95°C for 5 min. The generated cDNA was used to amplify the variable regions of the heavy and light chains of the expressed mAbs in two subsequent PCR reactions. These PCR reactions used Hotstar plus polymerase (Qiagen) in combination with NHP specific primers specifically designed for the cloning of rhesus macaque antibody V(D)J sequences [57](S3 Table). Both PCRs were done sequentially with 2 μl input cDNA and the following PCR reaction mix: forward and reverse primers (25 mM), 0.2 μl dNTPs (10 mM), Taq Hotstar plus polymerase (0.25 U) and 10x Hotstar plus buffer in a final reaction volume of 20 μl. PCR program used: 95°C 5 min; 50 cycles of 94°C, 30s; 55°C (PCR1)/60°C (PCR2), 30s; 72°C for 1 min; and a final step of 72°C for 10 min.

A final PCR reaction was performed with primers containing an overlapping region with the left or right arm of the restriction enzyme digested vector necessary for the ligation of the PCR product into corresponding Igγ1, Igκ and Igλ expression vectors [48]. Reaction mix contained 10x buffer, MgSO$_4$ (50 mM), dNTPs (10 mM), forward and reverse primer set and Platinum Taq polymerase (1 U) (All thermo Fisher Scientific) in a total volume of 25 μl. The PCR

program used was; 94˚C, 30s; then 25 cycles of 94˚C, 15 s; 60˚C, 30s; 68˚C, 1 min; final elonga-tion 10 min at 72˚C. Heavy and light chains were ligated into their corresponding expression vectors. First the expression vectors were digested with Thermo Scientific FastDigest restric-tion enzymes. The Igγ1 expression vector was digested using the FastDigest restriction enzymes BshTI and SalI. The Igκ expression vector was digested using FastDigest restriction enzymes BshtI and pf123II, whereas digestion of Igλ expression vector was done with FastDi-gest restriction enzymes BshtI and XhoI. Digestion of the expression vectors was confirmed using gel electrophoresis with an 1% agarose gel. Digested expression vectors were extracted from the agarose gel using the NucleoSpin Gel and PCR Clean-up kit and if necessary diluted to a concentration of 50 ng/μl.

The PCR products of the nested PCR were ligated into the corresponding vectors using the Gibson assembly method. The Gibson assembly reaction was performed with 1 μl PCR prod-uct, 1 μl digested Igγ1, Igκ and Igλ vector (50 ng/μl) and 2 μl 2x Gibson Assembly Master Mix (T5 exonuclease (0.2 U) (Epibio), Phusion polymerase (12.5 U) (New England Biolabs), Taq DNA ligase (2000 U) (New England Biolabs), Gibson reaction buffer (0.5 grams PEG-8000 (Sigma Life Sciences), 1 M Tris/HCl pH 7.5, 1 M MgCl2, 1M DTT, 100 mM dNTPs, 50 mM NAD (New England Biolabs), MQ). This mixture was incubated at 50˚C for 60 min. The plas-mid DNA was then transformed into α-select Escherichia coli (E.coli) (Bioline) using the heat shock method and grown on agar plates + ampicillin (100 μg/mL) overnight at 37˚C. Single colonies were taken the next day and grown overnight in 2 ml 2x LB medium + 2 μl ampicillin (100 mg/ml). Cloned plasmids were purified from E.coli using the standard protocol of the NucleoSpin Plasmid kit and were eluted with 50 μl of MilliQ.

### Phylogenetic tree generation

After the variable regions of the B cell receptors were cloned into their corresponding heavy and light chain human IgG1 expression vectors, they were sequenced with the BigDye Termi-nator v3.1 Cycle Sequencing Kit (ThermoFisher Scientific). Sequences were aligned in BioEdit with a ClustalW Multiple alignment and phylogenetic trees were generated with Cipres Sci-ence Gateway [58]. Phylogenetic trees were visualized with FigTree v1.4.4.

Each mAb lineage was named using the last two numbers of the NHP ID, preceded by RM for rhesus macaque, followed by a unique alphabetical lineage identifier. MAbs belonging to the same clonal family were indicated by an additional number e.g. RM46A1, RM46A2, RM46A3 are mAbs isolated from NHP 6446 belonging to the same clonal family.

### mAb expression

To produce the mAbs, 250 ml HEK293F cells (Invitrogen, cat no. R79009) were diluted to a cell density of 1 million cells/ml in FreeStyle Medium (Life Technologies) one hour before transfection. Heavy and light chain plasmids were co-transfected at a 1:1 ratio using PEImax (Polysciences Europe GmBH, Eppelheim, Germany) as the transfection reagent. After 5 days, the cells were spun down and the supernatant was filtered using 0.22 μm pore size Steritopes (Millipore, Amsterdam, The Netherlands). mAbs were purified using Protein G agarose (Pierce) affinity chromatography columns and eluted with 0.1 M glycine pH 2.5 into 1 M Tris PH 8. Buffer exchange into PBS was performed using Vivaspin20 centrifugal filters with 100 kDa MW cut-off (Sartorius, Göttingen, Germany).

### BG505 SOSIP and virus variants

For the 241 and 465 glycan knock-in BG505 SOSIP and virus variants, we substituted the ser-ine (BG505 S241N) and threonine (BG505 T465N), respectively, for an asparagine to create

the N(X)S/T motif necessary for glycosylation. To introduce a glycan at position 289, we substituted the proline for a serine at position 291 to produce the SOSIP (BG505 P291S) whereas for the virus the proline was substituted to a threonine (BG505 P291T). To remove the glycan at position 611, we substituted the corresponding asparagine to a glutamine (BG505 SOSIP N611Q) whereas for the virus the asparagine was substituted by an alanine (BG505 N611A). All variants were made using the quick change site directed mutagenesis kit (Agilent) following the manufacturer's instructions. We tailored the annealing temperatures for each primer set used. The virus variants were previously produced and are described in detail in Sanders *et al.*, 2015 and Klasse *et al.*, 2018[11,19].

## Neutralization assays

Neutralization assays were performed as described elsewhere [5, 59]. In brief, luciferase reporter TZMbl-cells (obtained through the NIH AIDS Reagent Program, Division of AIDS, NIAID, NIH from Dr. John C. Kappes, and Dr. Xiaoyun Wu) were seeded at ~17.000 cells in half-area 96-wells plates in 37.5 μl medium/well. On day one, mAbs with a start concentration of 100 μg/ml were 3-fold serial diluted and incubated with ~500 pg of virus for 1 hour at room temperature (RT). A positive control of virus without mAb, and a negative control of only TZMbl-cells were taken along for every sample. After incubation, DEAE (0.8 μg/ml) and san-quinvir (0.016 nM) in 12.5 μl total volume growth medium was added to the TZMBL-reporter cells. Subsequently, 50 μl mAb and virus mix was added to the TZMBL-cells and incubated for three days at 37˚C. After 3 days the luciferase activity was measured using the Bright-Glo (Promega) and GloMax Discover System. The inhibitory concentration ($IC_{50}$) were determined as the concentration of mAb were 50% of the virus was neutralized.

## Enzyme-linked immuno sorbent assays (ELISAs)

96-well half area plates were coated with *Galanthus nivalis* lectin (Vector Laboratories) at 20 μg/mL diluted in 0.1 M NaHC03 pH 8.6 (50 μl/well) overnight at 4˚C. Plates were washed with TBS and blocked with 1% (w/v) casein in PBS (Thermo Scientific) for 1 hour at RT. SOSIP trimers (2 μg/ml), diluted in 1% (w/v) casein in PBS (Thermo Scientific) were added (50 μl/well) and incubated for 2 hours at RT. Subsequently, 96-well plates were washed with TBS and incubated for 2 hours with 3-fold serial dilutions of mAbs (start concentration 1 μg/ml) in 1% (w/v) casein in PBS (Thermo Scientific), followed by three washes with TBS. A secondary antibody goat-anti-human conjugated to horseradish peroxidase (0.3 μg/mL) (Sera-Care) was added and incubated for 1 hour at RT, followed by five washes of TBS 0.05% Tween. Plates were developed with 50 μl/well of 0.1 M NaAc 0.1 M Citric acid with 0.01% $H_2O_2$ and 1% 3,3′,5,5′-Tetramethylbenzidine (TMB) (Sigma-Aldrich). The reaction was stopped with 25 μl 0.8 M $H_2SO_4$ and absorption was measured at 450 nanometer using the SPECTROestar nano spectrophotometer from BMG Labtech.

Competitive ELISAs were performed using the same reagents and minor adjustments to the protocol. In short, instead of 3-fold serial dilutions, the primary antibody was added at 10 μg/ml in triplicate. After 30 min a competitor mAb was added, without a washing step, at a concentration of 2x the $EC_{70}$ value and incubated for another 1.5 hours.

## Negative-stain electron microscopy

SOSIP/Fab complexes were made by mixing 15 μg SOSIP with a 6-fold per protomer molar excess for monoclonal Fab and allowed to incubate for 18 hrs at RT. Complex samples were SEC purified using a Superose 6 Increase 10/300 GL (GE Healthcare) column to remove excess Fab prior to EM grid preparation. Fractions containing the SOSIP/Fab complexes were pooled

and concentrated using 10 kDa Amicon spin concentrators (Millipore). Samples were diluted to 0.03 mg/mL in TBS (0.05 M Tris pH 7.4, 0.15 M NaCl) and adsorbed onto glow discharged carbon-coated Cu400 EM grids (Electron Microscopy Sciences) and blotted after 10 seconds. The grids were then stained with 3 μL of 2% (w/v) uranyl formate, immediately blotted, and stained again for 45 secs followed by a final blot. Image collection and data processing was performed as described previously on an FEI Talos microscope (1.98 Å/pixel; 72,000× magnification) with an electron dose of $\sim$25 electrons/$Å^2$ using Leginon [9,60]. 2D classification, 3D sorting and 3D refinement conducted using Relion v3.0 [61]. EM density maps were visualized using UCSF Chimera [62].

## Supporting information

**S1 Fig. Fluorescence-activated cell sorting strategies to select for BG505-specific B cells.** For the immunized NHPs, BG505-specific B cells were selected by fluorescence activated cell sorting using two differently labeled BG505 SOSIPs. The gating strategy of NHP ROp15 is shown here (left). In contrast, BG505-specific B cells from the BG505 SHIV-infected NHPs were sorted with a BG505 SOSIP and BG505 SOSIP S241N trimer. The gating strategy of NHP 6454 is shown here (right).
(PDF)

**S2 Fig. Neutralization of various BG505 virus variants.** To map the epitopes of the NAbs, we performed neutralization assays with a variety of virus variants. For each mAb, the $IC_{50}$ value against each pseudovirus variant relative to the BG505.T332N parental virus is given. This value is indicated as the relative inhibitory concentration 50 ($RIC_{50}$). If the NAbs were unable to neutralize a virus variant, the value indicates the $IC_{50}$ divided by the highest concentration that was used in the assay that gave no neutralization.
(PDF)

**S3 Fig. Binding competition of base-targeting mAbs with RM20G and RM19R.** (A) Competitive ELISA with base-targeting mAbs RM20G and RM19R isolated from BG505 SOSIP.664 immunized NHPs in a previous study. (B) Negative-stain electron microscopy 3D reconstruction of RM15A (purple) in complex with BG505 SOSIP (grey). RM15A was isolated from NHP ROp15.
(PDF)

**S4 Fig. Heterologous neutralization.** A selection of neutralizing antibodies were assessed for their neutralizing activity against a panel of heterologous viruses representing global HIV-1 diversity. None of the mAbs were able to neutralize any of these viruses indicated by the $IC_{50}$ value of >100 μg/mL.
(PDF)

**S1 Table. mAb characteristics.**
(PDF)

**S2 Table. Binding of all isolated mAbs to a variety of BG505 SOSIP variants by ELISA.**
(PDF)

**S3 Table. Overview of primers used in PCR amplification of the variable genes.**
(PDF)

## Author Contributions

**Conceptualization:** Jelle van Schooten, Marlies M. van Haaren, Laura E. McCoy, Rogier W. Sanders, George M. Shaw, Marit J. van Gils.

**Data curation:** Jelle van Schooten, Marlies M. van Haaren.

**Formal analysis:** Jelle van Schooten, Marlies M. van Haaren, Marit J. van Gils.

**Funding acquisition:** David C. Montefiori, Andrew B. Ward, Dennis R. Burton, John P. Moore, Rogier W. Sanders, Shane Crotty, George M. Shaw, Marit J. van Gils.

**Investigation:** Jelle van Schooten, Marlies M. van Haaren, Laura E. McCoy, Colin Havenar-Daughton, Christopher A. Cottrell, Judith A. Burger, Patricia van der Woude, Leanne C. Helgers, Ilhan Tomris, Celia C. Labranche, Marit J. van Gils.

**Methodology:** Jelle van Schooten, Marlies M. van Haaren, Laura E. McCoy, Colin Havenar-Daughton, Marit J. van Gils.

**Resources:** Hui Li, Colin Havenar-Daughton, John P. Moore, Shane Crotty, George M. Shaw.

**Supervision:** David C. Montefiori, Andrew B. Ward, Dennis R. Burton, John P. Moore, Rogier W. Sanders, Shane Crotty, George M. Shaw, Marit J. van Gils.

**Validation:** Jelle van Schooten, Marlies M. van Haaren, Marit J. van Gils.

**Visualization:** Jelle van Schooten, Marlies M. van Haaren.

**Writing – original draft:** Jelle van Schooten, Marlies M. van Haaren, Rogier W. Sanders, Marit J. van Gils.

**Writing – review & editing:** Jelle van Schooten, Marlies M. van Haaren, Laura E. McCoy, Dennis R. Burton, Rogier W. Sanders, George M. Shaw, Marit J. van Gils.

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
