## [Decision Letter · Decision Letter 0]

13 May 2021

Dear Mr. van Schooten,

Thank you very much for submitting your manuscript "Antibody responses induced by SHIV infection are more focused than those induced by soluble native HIV-1 envelope trimers in non-human primates" for consideration at PLOS Pathogens. As with all papers reviewed by the journal, your manuscript was reviewed by members of the editorial board and by several independent reviewers. The reviewers appreciated the attention to an important topic. Based on the reviews, we are likely to accept this manuscript for publication, providing that you modify the manuscript according to the review recommendations.

Sincerely,

Katie J Doores

Associate Editor

PLOS Pathogens

Susan Ross

Section Editor

PLOS Pathogens

Kasturi Haldar

Editor-in-Chief

PLOS Pathogens

orcid.org/0000-0001-5065-158X

Michael Malim

Editor-in-Chief

PLOS Pathogens

orcid.org/0000-0002-7699-2064

Reviewer Comments (if any, and for reference):

Reviewer's Responses to Questions

**Part I - Summary**

Reviewer #1: The manuscript by van Gils and colleagues compares side by side the antibody responses generated by BG505 SOSIP trimer immunization and BG505 SHIV infection in NHPs. The authors isolated mAbs from both animal populations and characterized their binding and neutralization profiles in detail. BG505 SHIV infection leads to a more pronounced clonal expansion and, interestingly, to a much lower extend of antibody diversity. BG505 SHIV infection produces mostly neutralizing antibodies targeting the 241/289 glycan hole and the V1 region. In contrast, BG505 SOSIP immunization generated antibodies that target more diverse sites including the glycan hole at N611, C3/V5, the gp41/gp120 interface and the V1 region. Notably, the V1 antibodies were the most potent neutralizing antibodies in both groups. However, neither V1 or other Abs neutralized heterologous viruses as reported in previous approaches and vaccine regimens. A substantial amount of non-neutralizing antibodies was also elicited targeting the base of the soluble SOSIP trimers as reported in previous studies using SOSIP-based soluble trimers.

Overall, the study design is novel and indicates some differences in antibody generation by soluble trimers compared to a natural infection setting within comparable time periods. The data is well analyzed and provides new clues to pursue further immunogen optimization such as diverging the immune response away from the variable V1 epitope towards more conserved known neutralizing Ab epitopes.

Reviewer #2: This paper entitled “Antibody responses induced by SHIV infection are more focused than those induced by soluble native HIV-1 envelope trimers in non-human primates” aims at comparing vaccine-induced antibody responses to BG505 SOSIP.664 immunogen to those induced by natural infection by a BG505 SHIV. They report that BG505 SHIV infection lead to more clonal expansion but less antibody diversity compared to the BG505 SOSIP immunization and that the SHIV elicited Mabs target less diverse epitopes. None are broad neutralizing antibodies, unfortunately.

This is a well written paper with a straightforward story.

**Part II – Major Issues: Key Experiments Required for Acceptance**

Reviewer #1: The BG505 SHIV probably does not contain any SOSIP modifications. The authors should explain whether the differences in antibody responses observed maybe due to the SOSIP stabilization modifications. Indeed the stabilized version may just present “better” defined epitopes and hence a broader Ab response.

Reviewer #2: However it is unclear what the advance is here and how useful it will be to advance HIV vaccine design.

Can the authors explain why they used different sorting strategy - does this not bias the result?

the authors could explain why they think there are differences in the Nab elicited - for some it is obvious but what about V1 vs 611 - maybe less accessible and the C3/V5…?

It could be interesting to understand the basis for the autologous neut (and not breadth) - higher resolution structure might provide that information.

**Part III – Minor Issues: Editorial and Data Presentation Modifications**

Reviewer #1: Line 493, the SOSP version should be indicated in the methods section.

The authors discuss strategies to focus the immune response. Within this context, the approach described by Dubrovskaya et al 2019 could be discussed as well.

Reviewer #2: It will be nice to remind the readers of the previous immunizations done and summarize the work that has been previously published (including SHIV infections).

The BG505 SHIV model and results have been published elsewhere and it is not surprising that no base directed Mabs will be elicited since it is indeed not present in the SHIV compared to the soluble BG505 SOSIP. The authors could mention this but they seem to be making a case which is likely not needed. It seems quite surprising that reports using BG505 SOSIP with glycans hiding the base of the trimer (for example) to prevent the elicitation of “base-directed Mab” have yet to be published.

Figure legend - more captions will be nice - animal ID or something.

Fig 2A - Please state the numbers of autologous nAb?

Although the analysis is interesting, why not show directly the neutralization and what are the properties of the Nab that correlate with neutralization

Between fig 1 and 2, which ones are the autologous N Mab?

Seems that the VH3 paired with Vl10 in 6454, VH3 paired with Vl1 in 43335 and vh5 paired with Vl1 in 6446…

Will be interesting to know if they correlated.

Assuming all the nab are shown in 2B.

Figure 3B, state the animal ID as well.

Color code the Fab vs Env

typo line 838 - longitudinal

PLOS authors have the option to publish the peer review history of their article (what does this mean?). If published, this will include your full peer review and any attached files.

Reviewer #1: No

Reviewer #2: No

Figure Files:

Data Requirements:

Reproducibility:

References:

---

## [Editor Report · Decision Letter 1]

21 Jun 2021

Dear Mr. van Schooten,

We are pleased to inform you that your manuscript 'Antibody responses induced by SHIV infection are more focused than those induced by soluble native HIV-1 envelope trimers in non-human primates' has been provisionally accepted for publication in PLOS Pathogens.

Best regards,

Katie J Doores

Associate Editor

PLOS Pathogens

Susan Ross

Section Editor

PLOS Pathogens

Kasturi Haldar

Editor-in-Chief

PLOS Pathogens

orcid.org/0000-0001-5065-158X

Michael Malim

Editor-in-Chief

PLOS Pathogens

orcid.org/0000-0002-7699-2064
---

## [Editor Report · Acceptance letter]

20 Aug 2021

Dear Mr. van Schooten,

We are delighted to inform you that your manuscript, "Antibody responses induced by SHIV infection are more focused than those induced by soluble native HIV-1 envelope trimers in non-human primates," has been formally accepted for publication in PLOS Pathogens.

Best regards,

Kasturi Haldar

Editor-in-Chief

PLOS Pathogens

orcid.org/0000-0001-5065-158X

Michael Malim

Editor-in-Chief

PLOS Pathogens

orcid.org/0000-0002-7699-2064